# The *CFTR* Gene Germline Heterozygous Pathogenic Variants in Russian Patients with Malignant Neoplasms and Healthy Carriers: 11,800 WGS Results

**DOI:** 10.3390/ijms24097940

**Published:** 2023-04-27

**Authors:** Maria Makarova, Marina Nemtsova, Anastasiia Danishevich, Denis Chernevskiy, Maxim Belenikin, Anastasiia Krinitsina, Elena Baranova, Olesya Sagaydak, Maria Vorontsova, Igor Khatkov, Lyudmila Zhukova, Natalia Bodunova, Sergey Nikolaev, Mariya Byakhova, Anna Semenova, Vsevolod Galkin, Saida Gadzhieva

**Affiliations:** 1LLC Evogen, 115191 Moscow, Russia; makarova@evogenlab.ru (M.M.);; 2Federal State Budgetary Institution Russian Scientific Center of Roentgenoradiology of the Ministry of Healthcare of the Russian Federation, 117997 Moscow, Russia; 3Research Centre for Medical Genetics of N.P. Bochkov, 115522 Moscow, Russia; 4Federal State Autonomous Educational Institution of Higher Education I.M. Sechenov of the Ministry of Health of Russian Federation, 119991 Moscow, Russia; 5SBHI Moscow Clinical Scientific Center Named after Loginov MHD, 111123 Moscow, Russia; 6Academy of Continuing Professional Education of the Ministry of Health of Russian Federation, 125993 Moscow, Russia; 7The National Medical Research Center for Endocrinology, 117292 Moscow, Russia; 8City Clinical Oncological Hospital No. 1, Moscow Department of Healthcare, 117152 Moscow, Russia

**Keywords:** cystic fibrosis, *CFTR*, hereditary cancer predisposition syndrome, the frequency of heterozygous carriage, Russian cohort of patients

## Abstract

More than 275 million people in the world are carriers of a heterozygous mutation of the *CFTR* gene, associated with cystic fibrosis, the most common autosomal recessive disease among Caucasians. Some recent studies assessed the association between carriers of *CFTR* variants and some pathologies, including cancer risk. The aim of this study is to analyze the landscape of germline pathogenic heterozygous *CFTR* variants in patients with diagnosed malignant neoplasms. For the first time in Russia, we evaluated the frequency of *CFTR* pathogenic variants by whole-genome sequencing in 1800 patients with cancer and compared this with frequencies of *CFTR* variants in the control group (1825 people) adjusted for age and 10,000 healthy individuals. In the issue, 47 out of 1800 patients (2.6%) were carriers of *CFTR* pathogenic genetic variants: 0.028 (42/1525) (2.8%) among breast cancer patients, 0.017 (3/181) (1.7%) among colorectal cancer patients and 0.021 (2/94) (2.1%) among ovarian cancer patients. Pathogenic *CFTR* variants were found in 52/1825 cases (2.85%) in the control group and 221 (2.21%) in 10,000 healthy individuals. Based on the results of the comparison, there was no significant difference in the frequency and distribution of pathogenic variants of the *CFTR* gene, which is probably due to the study limitations. Obviously, additional studies are needed to assess the clinical significance of the heterozygous carriage of *CFTR* pathogenic variants in the development of various pathologies in the future, particularly cancer.

## 1. Introduction

Cystic fibrosis (OMIM #219700) (CF) is the most common autosomal recessive disease among Caucasians (1:2000–1:10,000 newborns). CF develops in the presence of two inactivating germline mutations in the *CFTR* gene encoding the protein of the same name, CFTR (cystic fibrosis transmembrane conductance regulator).

Proteins of the chloride channel superfamily, including the CFTR protein, play a key role in gastrointestinal tract (GIT) homeostasis. Their functions include osmoregulation, ion transport across the epithelium, cellular metabolism, cellular autophagy, cell migration, mucus secretion, innate and adaptive immune responses, intercellular interactions, membrane potential, oxidative phosphorylation, inflammation, microbiome composition, cell pH, and apoptosis [1]. CFTR is expressed by cells of the entire GIT with a down-gradient expression from the proximal (duodenum) to the distal part (ileum) [2]. CFTR dysfunction in the GIT is accompanied by a decrease in intestinal pH, an increase in mucus viscosity, and an impaired immune response, leading to local GIT inflammation. What is more, the frequent systemic antibiotic therapy, required by patients with CF, leads to a significant change in the intestinal microflora, enhancing the pro-inflammatory effect. Chronic inflammation caused by an impaired CFTR and long-term antibiotic therapy in combination with environmental factors could contribute to carcinogenesis [1,3].

The development of effective lung disease treatment for people with CF has significantly increased the life expectancy of patients, exceeding 46 years nowadays [4]. An increased life expectancy allowed for the initiation of a 20-year epidemiological study, comparing the incidence of malignancies among patients with CF with the general population. The study showed an increase in the incidence of malignant neoplasms (MNs) of the GIT, testicles and lymphoid leukemia among patients with CF [5]. A meta-analysis of six population-based studies over the years (more than 99,000 patients) confirmed the high risk of developing GIT cancer in CF patients [6]. The risk of developing colorectal cancer (CRC) was one of the highest: according to various estimates, it is 5–10 times higher than in the general population [2,5,7,8]. According to the accumulated data, colonoscopy is recommended as a screening for CRC in CF patients aged 40 years and older, and in the presence of a weakened immune system (in organ transplant recipients), aged 30 years and older [9].

Until recently, the presence of one wild-type allele of the CFTR gene was considered to be sufficient for the proper functioning of the gene, and heterozygous carriers of pathogenic variants in *CFTR* were not considered to be at an increased risk of pathological conditions [10]. According to some estimates, more than 275 million people in the world are carriers of a heterozygous mutation of the *CFTR* gene, with Caucasians being the majority [4]. Nevertheless, some studies have found some pathologies typical to CF among the carriers: congenital bilateral absence of the vas deferens, sinusitis, pancreatitis, bronchiectasis, mycobacterial infections, asthma and malignancies [11]. In Russia, analysis of the *CFTR* gene for heterozygous carrier identification is recommended only at the stage of pregnancy planning, as well as for male patients with several forms of infertility. To date, the frequency and spectrum of pathogenic *CFTR* variants have not been assessed among Russian patients with diagnosed MNs. The aim of this study is to analyze the landscape of germline pathogenic heterozygous *CFTR* variants in patients with diagnosed malignant neoplasms.

## 2. Results

Blood samples from the 1800 patient group (PG), 1825 control group (CG) and 10,000 healthy individual group (HIG) were subjected to WGS (Table 1).

PG whole-genome sequencing (WGS) results: 47/1800 patients (2.6%) were carriers of *CFTR* pathogenic genetic variants, including 45 women and two men. The mean female carrier age was 44.5 ± 7.7 years, the mean male carrier age was 66 ± 3 years. The average age of patients can be explained by the inclusion criteria in the study.

The largest proportion of the *CFTR* genetic alterations were found in women with breast cancer (BC): 41 out of 47 carriers (87%).

The distribution by sex and diagnosis is explained by the majority of BC patients included in the study, i.e., 1514 (84%).

In 47 carriers, 47 *CFTR* pathogenic variants were identified. The landscape of the identified genetic variants is described in Table 2.

Among 47 carriers, 40 (85%) have pathogenic genetic variants that are included in the core panel for CF diagnosis (microarray) [12], that is comprised of 28 most frequent pathogenic variants of the *CFTR* gene in Russia, which is used for comparison. It should be emphasized that seven out of 47 (15%) identified genetic variants are not included in this panel.

The distribution of the identified *CFTR* pathogenic variants, depending on the localization of the primary tumor, is as follows: in the cohort of BC patients, the frequency of pathogenic variants was 0.028 (42/1525) (2.8%), in CRC patients 0.017 (3/181) (1.7%) and ovarian cancer (OC) 0.021 (2/94) (2.1%).

The p.Phe508del (chr7:117559592_117559594del, NM_000492.4:c.1520_1522del, rs113993960) variant was the most prevalent in our study, with 32/47 (68%) identified variants. p.Phe508del is part of the core panel for “frequent” mutations and is recognized as the most common (up to 70% of cases) cause of CF in both Russia and the world in homozygous and compound-heterozygous forms. The variant frequency is 0.00789 (0.7897%) in gnomAD genomes v. 3.1.1.

According to the results of WGS of the control group (CG), clinically significant variants were found in 52/1825 cases (2.85%). The landscape of the identified variants is presented in Table 3.

Pathogenic genetic variants of 45/52 carriers are included in the core panel of most frequent *CFTR* genetic alterations in Russia and 7/52 (14%) identified variants are not included in this panel.

As in the PG, in the CG, the variant p.Phe508del (chr7:117559592_117559594del, NM_000492.4:c.1520_1522del, rs113993960) was the most frequent genetic alteration, with 34/52 variants (65%).

Additionally, this study analyzed WGS data from 10,000 healthy people without cystic fibrosis and without malignant neoplasms. The analysis found 221 (2.21%) individuals heterozygous for *CFTR* pathogenic variants. The mutation landscape is presented in Table 4.

## 3. Discussion

For the first time in Russia, a full-scale molecular genetic study of the *CFTR* gene was carried out in patients with malignant neoplasms. WGS of 1800 Russian patients with malignant neoplasms, 1825 people of the control group and 10,000 healthy people in total was carried out. The use of WGS facilitated the analysis of genetic variants in known cancer-associated genes, as well as the evaluation of the frequency and distribution of the *CFTR* genetic variants with their subsequent comparison with the distribution in the CG. In the current study, we identified pathogenic variants that are not included in the conventionally used core panel of frequent *CFTR* variants: seven variants in the PG and seven variants in the CG, which indicates more effective testing compared to the core panel of most frequent *CFTR* genetic alterations in Russia.

This is the first widescale frequency study of the *CFTR* pathogenic variants carriage in Russia.

The previous large study was performed in 2020 and consisted of an analysis of the 60 *CFTR* pathogenic variants among 642 healthy people. The researchers obtained similar results, whereby 23 heterozygous carriers were identified; the frequency was 3.58% (95% CI: 2.28–5.33%). The Phe508del (rs113993960) variant, as expected, became the leader, with a frequency of 2.02% [13].

Recently, there has been an increased interest in the world regarding the influence of pathogenic variants in the genes that determine the development of hereditary pathology in malignant neoplasms. A large population-based study involving about 500,000 people assessed the association between carriers of a mutation in the *CFTR* gene, Phe508del, and the risk of developing 54 types of cancer [8]. Compared with the control group, an increased frequency of the pathogenic *CFTR* variant p.Phe508del was found in individuals with colorectal cancer (OR 1.17 (95% CI 1.02–1.32, *p* = 0.02)), gallbladder and biliary tract cancer (OR 1.92 (95% CI 1.20–2.91, *p* = 0.004)), thyroid cancer (OR 1.47 (95% CI 0.99–2.08, *p* = 0.04)) and non-Hodgkin’s lymphoma (OR 1.32 (95% CI 1.04–1.65, *p* = 0.02)).

The present study additionally evaluated the frequency of pathogenic variants of the *CFTR* gene in 1525 patients with breast cancer, 181 patients with CRC and 94 patients with OC. The frequencies of pathogenic *CFTR* variants were compared, with the frequencies of these variants in the control group adjusted for age. Based on the results of the comparison, there was no significant difference in the frequency and distribution of pathogenic variants of the *CFTR* gene (PG:OR 1.181 (95% CI 0.859–1.625, *p* = 0.305), BC:OR 1.225 (95% CI 0.874–1.718, *p* = 0.197), CRC:OR 0.750 (95% CI 0.238–2.366, *p* = 0.623), OC:OR 0.963 (95% CI 0.243–3.821, *p* = 0.958)), which is probably due to either an insufficient number of individuals in both patient and control cohorts, or features of the cohort. In the cohort of the current study, most frequent patients had a breast cancer diagnosis.

It is known that a deficiency in the expression of the CFTR protein is associated with an increased risk of sporadic colorectal cancer, but the mechanism of this effect is not fully resolved. Currently, there are several hypotheses that might potentially explain the association of CFTR deficiency with an increased risk of colorectal cancer. One is the effect of CFTR on intestinal stem cells, which are the main source of CRC progenitor cells. Li et al. reported that mouse stem cells with the *CFTR* genetic variant (F508del) are prone to the development of teratomas and activation of genes that mediate the epithelial–mesenchymal transition and are involved in proliferation and migration [14].

Additionally, several studies demonstrate that *CFTR* is associated with the regulation of signaling of the pathological Wnt/β-catenin pathway. Dysregulation of the Wnt/β-catenin signaling pathway is involved in the development of up to 90% of cases of CRC in humans, contributing to both the early onset of the tumor and the progression of invasive CRC [15]. However, given that CFTR deficiency has been found in a variety of tumor types that occur independently of Wnt/β-catenin signaling, it is possible that the CFTR function as a tumor suppressor extends beyond the Wnt/catenin pathway.

Another important mechanism that may lead to the development of CRC is the effect of CFTR on the processes that are involved in the maintenance of tissue homeostasis in the intestine. These include the composition of the intestinal microflora, maintaining the main barriers that protect the unicellular epithelial layer of the large intestine from bacterial invasion, maintaining homeostasis of the innate and adaptive immune response. Clinical manifestations of CF in the gastrointestinal tract, including inflammation and obstruction, are associated with the dysregulation of these processes, which creates favorable conditions for the development of cancer [2].

Currently, there is evidence that a violation of the expression of the *CFTR* gene is related to the development of cancer not only of the gastrointestinal tract, but also of other types of cancer: head and neck, non-small cell lung cancer, bladder, liver (hepatocellular cancer) and breast cancers [16,17,18,19].

The role of *CFTR* in the pathogenesis of various types of malignancies is not limited to genetic variants, but also manifests itself through epigenetic modifications [20]. It has previously been shown that the *CFTR* gene promoter is hypermethylated in tumor tissue in breast cancer, while the level of CFTR messenger RNA decreases. Treatment of breast cancer cells with decitabine (10 µM), which removes hypermethylation, leads to the restoration of *CFTR* mRNA expression. Hypermethylation of *CFTR* in patients with breast cancer is associated with the development of invasive carcinoma. In addition, low levels of the CFTR protein correlate with a poor survival of patients with breast cancer [21].

Possibly, the significance of heterozygous carriage of clinically significant genetic variants in the *CFTR* gene in the development of various types of pathology is no longer as harmless as it used to be perceived and will potentially be revised in the future.

The *CFTR* pathogenic variants carriage was analyzed in patients with malignant neoplasms (1800 samples), in the control group (1825 samples) and among healthy people (10,000 samples). The Phe508del variant is the most frequent across all groups. WGS revealed rare variants; however, in a small number and their association with hereditary neoplasms has not been studied. There was no statistically significant difference in the variant frequency among different groups. However, it is important to emphasize that there are international studies on the association of *CFTR* and colorectal cancer, so in future studies, an improved male/female ratio and focus on gastrointestinal tumors, with a previously reported increased ratio in CF, should be aimed for. For these studies, the NGS panel can be used, which includes cancer-associated genes and the *CFTR* gene.

The use of WGS in the future will make it possible to assess the full range of molecular genetic alterations of the *CFTR* gene, including extended deletions/insertions, as well as clinically significant variants in the intron and regulatory regions of the gene. Perhaps, such an approach will increase the interest of the scientific community regarding this problem, which will facilitate the evaluation of its association with various CF-associated pathological conditions, as well as with the process of carcinogenesis.

## 4. Materials and Methods

A large-scale scientific research project was implemented in Moscow (Russia) in 2021–2022 to identify hereditary cancer syndromes (HCS) by WGS in the PG with CRC, BC and/or OC. Within the framework of the project, an additional study of the carrier and landscape of pathogenic *CFTR* variants in this cohort was carried out. The obtained results were compared with the CG, that was adjusted for age and did not include patients with diagnosed oncological diseases at the time of the study.

*CFTR* gene variants identified among the PG, CG and 10,000 healthy people without cystic fibrosis and without malignant neoplasms were compared with the core panel (Table 5).

**Recruitment criteria.** Blood samples from the 1800 PG, 1825 CG and 10,000 HIG underwent WGS.

Inclusion criteria for PG were as follows: individuals aged 18 years and older with BC and/or OC diagnosis in women, BC in men and CRC in combination with the patient’s age <50 years, and/or the presence of multiple tumors, and/or the presence of cancer cases in family history. The PG blood samples were collected from 6 public oncology hospitals in Moscow.

Among 1800 patients included in the PG, there were 1699 women (94.4%) and 101 men (5.6%). The average age of the patients was 45.85 ± 9.06 years, ranging from 18 to 83 years. In 1389 cases (77%), the age of the patients was <50 years and in 411 age was ≥50 years.

The WGS analysis results of 1825 samples from the CG, adjusted for sex and age, were obtained.

The WGS analysis results of 10,000 healthy people without cystic fibrosis and without malignant neoplasms were obtained. WGS was performed as part of health insurance. This cohort included people from 20 to 76 years old (mean, 40.2 ± 7.7), of which 4837 (48.4%) were men, the mean age was 39.5 ± 7.6 and 5163 (51.63%) were women, mean age 40.8 ± 7.7 years.

WGS was performed in the LLC Evogen laboratory.

As part of a WGS analysis, the identification of clinically significant genetic variants associated with either HCS or other hereditary diseases with similar phenotypic manifestations, was performed.

**DNA isolation.** DNA was obtained from the blood buffy coat. DNA isolation was performed from 200 µl peripheral blood using the QIAamp DNA blood Mini Kit (QIAGEN, Hilden, Germany), according to the manufacturer’s standard protocol. Qualitative and quantitative assessments were carried out spectrophotometrically (NanoDrop 8000, and Denovix DS-11, Waltham, MA, USA) and fluorometrically (Qubit4, Qubit flex and Denovix QFX, Waltham, MA, USA), respectively.

**WGS.** WGS was performed using NGS high-throughput sequencers DNBseq-T7 and DNBseq-G400 (MGI, Shenzhen, China), using a PCR-free enzymatic shearing protocol for library preparation (MGIEasy FS PCR-Free DNA Library Prep Kit (MGI, Shenzhen, China)). All the subsequent stages, including PE150 (paired-end 2 × 150 bp) sequencing, were carried out in accordance with the manufacturer’s standard protocols. The average sequencing depth was 30×. The identification of the genetic variants was carried out using MegaBOLT bioinformatics analysis accelerators (MGI, Shenzhen, China). The average number of identified genetic alterations was about 4.5 million per sample. During result analysis, special expert attention was carried out primarily for cancer-associated genes for patients with malignant tumors. The gnomAD database was used to estimate the population frequencies of the identified variants [22].

During research for genetic variant annotation, a number of databases were used: OMIM (Online Mendelian Inheritance in Man) [23], Cancer Genome Interpreter (Identification of therapeutically actionable genomic alterations in tumors) [24], My Cancer Genome [25], NCBI (National Center for Biotechnology Information) databases [26], VarSome (The Human Genomics Community) [27], ACMG (American College of Medical Genetics and Genomics), as well as medical expert information and data from scientific research literature. Since the main focus of this article is the analysis of the structure of variants in the *CFTR* gene, specialized sites and resources were also used, e.g., cftr.iurc.montp.inserm.fr [28]. Pathogenic, likely pathogenic variants and variants of uncertain clinical significance (VUS) in onco-associated genes and the *CFTR* gene were analyzed during the assessment of clinical significance of identified genetic alterations. WGS with a standard sequencing depth of ~30× is not intended for the detection of certain types of genetic variants: structural variants in chromosomes (inversions, translocations, copy number variants), polyploidy, aneuploidy, repetitive elements; variants in genes with pseudogenes; epigenetic changes. For this reason, large deletions of the *CFTR* gene were not analyzed in this study, but such a study is planned in future.

**Statistical Analysis.** The relative cancer risk was assessed using an odds ratio (OR) with a 95% confidence interval (CI). Statistical analysis was carried out using the statistical program STATISTICA version 13.5.0. *p*-values < 0.05 were considered statistically significant for all comparisons.

## Figures and Tables

**Table 1 ijms-24-07940-t001:** Distribution of identified variants of the *CFTR* gene in the PG, CG and HIG.

No.	Primary Tumor Localization	Number of Patients with *CFTR* Pathogenic Genetic Variants	Frequency
1.	Patient group (PG)	47/1800	0.026 (2.6%)
2.	Breast cancer	42/1525	0.027 (2.27%)
3.	Ovarian cancer	2/94	0.0212 (2.12%)
4.	Colorectal cancer	3/181	0.016 (1.6%)
5.	Control group (CG)	52/1825	0.0285 (2.85%)
6.	Healthy individual group (HIG)	221/10,000	0.0221 (2.21%)

**Table 2 ijms-24-07940-t002:** The spectrum of identified genetic variants of the *CFTR* gene in the PG.

No.	Genetic Variants of the *CFTR* Gene (NM_000492.4, HG38)	rsID	Number of Carriers	Included in Core Panel
**1.**	c.1520_1522del(p.Phe508del)	rs113993960	32	Yes
**2.**	c.350G>A(p.Arg117His)	rs78655421	3	No
**3.**	c.3909C>G(p.Asn1303Lys)	rs80034486	2	Yes
**4.**	c.1624G>T(p.Gly542Ter)	rs113993959	1	Yes
**5.**	c.274G>A(p.Glu92Lys)	rs121908751	1	Yes
**6.**	c.3691del(p.Ser1231ProfsTer4)	rs77035409	1	Yes
**7.**	c.1545_1546del(p.Tyr515_Arg516delinsTer)	rs121908776	1	Yes
**8.**	c.1585-1G>A	rs76713772	1	Yes
**9.**	c.3846G>A(p.Trp1282Ter)	rs77010898	1	Yes
**10.**	c.328G>C(p.Asp110His)	rs113993958	1	No
**11.**	c.617T>G(p.Leu206Trp)	rs121908752	1	No
**12.**	c.442del(p.Ile148LeufsTer5)	rs121908770	1	No
**13.**	c.3841C>T(p.Gln1281Ter)	rs397508615	1	No
**Total**			**47**	**40**

**Table 3 ijms-24-07940-t003:** Spectrum of identified genetic variants of the *CFTR* gene in the CG.

No.	Genetic Variants of the *CFTR* Gene (NM_000492.4, HG38)	rsID	Number of Carriers	Included in the Core Panel
1.	c.1521_1523del(p.Phe508del)	rs113993960	34	Yes
2.	c.350G>A(p.Arg117His)	rs78655421	2	No
3.	c.3883del(p.Ile1295PhefsTer33)	rs397508630	1	No
4.	c.3909C>G(p.Asn1303Lys)	rs80034486	1	Yes
5.	c.3691del(p.Ser1231ProfsTer4)	rs77035409	1	Yes
6.	c.262_263del(p.Leu88IlefsTer22)	rs121908769	1	Yes
7.	c.4004T>C(p.Leu1335Pro)	rs397508658	2	No
8.	c.2052dup(p.Gln685ThrfsTer4)	rs121908746	3	Yes
9.	c.1397C>G(p.Ser466Ter)	rs121908805	1	No
10.	c.3587C>G(p.Ser1196Ter)	rs121908763	1	Yes
11.	c.274G>A(p.Glu92Lys)	rs121908751	2	Yes
12.	c.2195T>G(p.Leu732Ter)	rs397508350	1	No
13.	c.1040G>A(p.Arg347His)	rs77932196	1	Yes
14.	c.2012delT(p.Leu671Ter)	rs121908812	1	Yes
**Total**			**52**	**45**

**Table 4 ijms-24-07940-t004:** The *CFTR* mutation landscape identified through WGS in 10,000 healthy people.

No.	Genetic Variants of the *CFTR* Gene (NM_000492.4, HG38)	rsID	Number of Carriers	Included in the Core Panel
1.	c.1521_1523del(p.Phe508del)	rs113993960	131	Yes
2.	c.350G>A(p.Arg117His)	rs78655421	15	No
3.	c.413_415dup(p.Leu138dup)	rs397508686	7	Yes
4.	c.349C>T(p.Arg117Cys)	rs77834169	6	No
5.	c.2052dup(p.Gln685ThrfsTer4)	rs121908746	5	Yes
6.	c.274G>A(p.Glu92Lys)	rs121908751	5	Yes
7.	c.3909C>G(p.Asn1303Lys)	rs80034486	5	Yes
8.	c.4004T>C(p.Leu1335Pro)	rs397508658	4	No
9.	c.3587C>G(p.Ser1196Ter)	rs121908763	3	Yes
10.	c.1397C>G(p.Ser466Ter)	rs121908805	3	No
11.	c.2012delT(p.Leu671Ter)	rs121908812	3	Yes
12.	c.4426C>T(p.Gln1476Ter)	rs374705585	3	No
13.	c.3929G>A(p.Trp1310Ter)	rs397508645	3	No
14.	c.262_263del(p.Leu88IlefsTer22)	rs121908769	2	Yes
15.	c.1545_1546del(p.Tyr515Ter)	rs121908776	2	Yes
16.	c.2834C>T(p.Ser945Leu)	rs397508442	2	No
17.	c.1657C>T(p.Arg553Ter)	rs74597325	2	Yes
18.	c.3691del(p.Ser1231ProfsTer4)	rs77035409	2	Yes
19.	c.1040G>A(p.Arg347His)	rs77932196	2	No
20.	c.328G>C(p.Asp110His)	rs113993958	1	No
21.	c.1624G>T(p.Gly542Ter)	rs113993959	1	Yes
22.	c.2374C>T(p.Arg792Ter)	rs145449046	1	No
23.	c.1911del(p.Gln637HisfsTer26)	rs1554389296	1	No
24.	c.1510G>T(p.Glu504Ter)	rs397508223	1	No
25.	c.1A>G(p.Met1Val)	rs397508328	1	No
26.	c.2195T>G(p.Leu732Ter)	rs397508350	1	No
27.	c.2491G>T(p.Glu831Ter)	rs397508387	1	No
28.	c.2589_2599del(p.Ile864SerfsTer28)	rs397508400	1	No
29.	c.3475T>C(p.Ser1159Pro)	rs397508572	1	No
30.	c.3883del(p.Ile1295PhefsTer33)	rs397508630	1	No
31.	c.4300_4301dup(p.Ser1435GlyfsTer14)	rs397508709	1	No
32.	c.1652G>A(p.Gly551Asp)	rs75527207	1	Yes
33.	c.3846G>A(p.Trp1282Ter)	rs77010898	1	Yes
34.	c.1040G>C(p.Arg347Pro)	rs77932196	1	Yes
35.	c.3472C>T(p.Arg1158Ter)	rs79850223	1	No
**Total**			221	171 (77.4%)

**Table 5 ijms-24-07940-t005:** One of the core panels of *CFTR* variants (microarray) used in Russia [12].

No.	Chromosome/ Position	Traditional Name	Genetic Variants of the *CFTR* Gene (NM_000492.4)	rsID
1	chr7:117559591	F508del	c.1520_1522del(p.Phe508del)	rs113993960
2	chr7:117652877	N1303K	c.3909C>G(p.Asn1303Lys)	rs80034486
3	chr7:117587778	G542X	c.1624G>T(p.Gly542Ter)	rs113993959
4	chr7:117627742	3821delT	c.3691del(p.Ser1231ProfsTer4)	rs77035409
5	chr7:117587738	1717-1G>A	c.1585-1G>A	rs76713772
6	chr7:117642566	W1282X	c.3846G>A(p.Trp1282Ter)	rs77010898
7	chr7:117559613	1677delTA	c.1545_1546del(p.Tyr515_Arg516delinsTer)	rs121908776
8	chr7:117530899	E92K	c.274G>A(p.Glu92Lys)	rs121908751
9		Dele2-3	c.54-5940_273+10250del21kb(p.Ser18ArgfsX16)	-
10	chr7:117540270	R347H	c.1040G>A(p.Arg347His)	rs77932196
11	chr7:117592212	2184insA	c.2052dup(p.Gln685fs)	rs121908746
12	chr7:117627651	3732delA	c.3600del(p.Asp1201MetfsTer10)	-
13	chr7:117509123	G85E	c.254G>A(p.Gly85Glu)	rs75961395
14	chr7:117540176	1078delT	c.948del(p.Phe316fs)	rs75528968
15	chr7:117587806	G551D	c.1652G>A(p.Gly551Asp)	rs75527207
16	chr7:117592218	2183AA-G	c.2051_2052delinsG(p.Lys684fs)	rs121908799
17	chr7:117531115	621+1G>T	c.489+1G>T	rs78756941
18	chr7:117559587	I507del	c.1516ATC(p.Ile507del)	rs121908745
19	chr7:117587811	R553X	c.1657C>T(p.Arg553Ter)	rs74597325
20	chr7:117602868	2789+5G>A	c.2657+5G>A	rs80224560
21	chr7:117639961	3849+10kbC>T	c.3718-2477C>T	rs75039782
22	chr7:117540230	R334W	c.1000C>T(p.Arg334Trp)	rs121909011
23	chr7:117627537	R1162X	c.3484C>T(p.Arg1162Ter)	rs74767530
24	chr7:117540270	R347P	c.1040G>C(p.Arg347Pro)	rs77932196
25	chr7:117592178	2143delT	c.2012del(p.Ser670_Leu671insTer)	rs121908812
26	chr7:117627640	S1196X	c.3587C>G(p.Ser1196Ter)	rs121908763
27	chr7:117509128	394delTT	c.262_263del(p.Leu88fs)	rs121908769
28	chr7:117531036	L138ins	c.413_415dup(p.Leu138dup)	rs397508686

## Data Availability

The data are not publicly available due to restrictions, as they contain information that could compromise the privacy of research participants. Requests to access the additional data should be addressed to the following email: belenikin@evogenlab.ru.

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
