# Peer review of "The CFTR Gene Germline Heterozygous Pathogenic Variants in Russian Patients with Malignant Neoplasms and Healthy Carriers: 11,800 WGS Results"

_ijms, 2023, doi:10.3390/ijms24097940_

Round 1
Reviewer 1 Report
The authors present evidence of the first widescale WGS analysis of greater than 10,000 CF carriers and matched controls in a Russian population with malignant neoplasms. The authors found no difference in the CF carrier rate when comparing the control to cancer patients for several types of neoplasms (breast, colorectal, and ovarian).
This paper only contains clinical DNA sequencing data and thus is not fit for publication in IJMS (instructions to authors, “Papers that only contain clinical trials/data are not acceptable for IJMS.”). A sister mdpi journal such as “Genes“ is more suitable for this type of manuscript.
Suggestion major and minor changes to manuscript:
Line 75 – according to the accumulated data, colonoscopy is …… (add coma after data)
Line 87 & 236 – change “carriage” to carrier
Line 98 – Table one, the “M” and “W” do not line with the cancer types, please identify each cancer type listed in table as male or female
Line 107-108 – table title spans two lines and “in the PG” is bolded, reformat to make title on line 107 and remove bold from “in the PG”
Line 115 – duplication of “on the”, remove one occurrence
Line CRC, BC, and OC should be defined upon first use and not only in materials and methods
Line 157 – “which indicates more effective testing.” More effective compared to what?
Line 159 – “that determine the development of hereditary pathology 159 on the development of malignant neoplasms.” Maybe recast to state “that determine the development of hereditary pathology in malignant neoplasms”
In discussion, provide comparison of results with studies done in other countries or populations.
Line 169 – missing comma in 1 525 should be 1,525
Line 292 – typo “standart” change to “standard”
Line 232 - Materials and methods section is difficult to read. Suggest that subsections be created with subtitles for the different techniques. For example, collection of samples, DNA isolation, database analysis, etc.
Line 297 – states descriptive statistics were used from Excel. Please provide more details. What statistics used?
Line 314 - Authors need to justify in their data availability statement why data is not deposited in public database or made available to research community. Written statement could “The data are not publicly available due to [insert reason here]”
Author Response
Thank you very much for your revision.
The authors present evidence of the first widescale WGS analysis of greater than 10,000 CF carriers and matched controls in a Russian population with malignant neoplasms. The authors found no difference in the CF carrier rate when comparing the control to cancer patients for several types of neoplasms (breast, colorectal, and ovarian).
This paper only contains clinical DNA sequencing data and thus is not fit for publication in IJMS (instructions to authors, “Papers that only contain clinical trials/data are not acceptable for IJMS.”). A sister mdpi journal such as “Genes“ is more suitable for this type of manuscript.
Suggestion major and minor changes to manuscript:
Line 75 – according to the accumulated data, colonoscopy is …… (add coma after data)
Done.
Line 87 & 236 – change “carriage” to carrier
Done.
Line 98 – Table one, the “M” and “W” do not line with the cancer types, please identify each cancer type listed in table as male or female
We have replaced Tab.1 with a more informative one.
Line 107-108 – table title spans two lines and “in the PG” is bolded, reformat to make title on line 107 and remove bold from “in the PG”
Done.
Line 115 – duplication of “on the”, remove one occurrence
Done.
Line CRC, BC, and OC should be defined upon first use and not only in materials and methods
Done.
Line 157 – “which indicates more effective testing.” More effective compared to what?
Added «more effective testing compared to the core panel of most frequent CFTR genetic alterations in Russia»
Line 159 – “that determine the development of hereditary pathology 159 on the development of malignant neoplasms.” Maybe recast to state “that determine the development of hereditary pathology in malignant neoplasms”
Done.
In discussion, provide comparison of results with studies done in other countries or populations.
Discussion includes the link to population-based study (Shi, Z.; Wei, J.; Na, R.; Resurreccion, W.K.; Zheng, S.L.; Hulick, P.J.; Helfand, B.T.; Talamonti, M.S.; Xu, J. Cystic fibrosis F508del carriers and cancer risk: Results from the UK Biobank. International journal of cancer. 2021, 148(7), 1658–1664.).
We also added the link to the previous study in Russia in 2020 (Kiseleva A, Klimushina M, Sotnikova E, Skirko O, Divashuk M, Kurilova O, Ershova A, Khlebus E, Zharikova A, Efimova I, Pokrovskaya M, Slominsky PA, Shalnova S, Meshkov A, Drapkina O. Cystic Fibrosis Polymorphic Variants in a Russian Population. Pharmgenomics Pers Med. 2020 Dec 1;13:679-686. doi: 10.2147/PGPM.S278806).
Line 169 – missing comma in 1 525 should be 1,525
Done.
Line 292 – typo “standart” change to “standard”
Done.
Line 232 - Materials and methods section is difficult to read. Suggest that subsections be created with subtitles for the different techniques. For example, collection of samples, DNA isolation, database analysis, etc.
Done.
Line 297 – states descriptive statistics were used from Excel. Please provide more details. What statistics used?
Added.
Line 314 - Authors need to justify in their data availability statement why data is not deposited in public database or made available to research community. Written statement could “The data are not publicly available due to [insert reason here]”
Done.
Reviewer 2 Report
I have with interest read this study. See below some suggestions for clarification.
Abstract: add that in addition to cancer and control group, WGS was performed in 10000 healthy individuals, add results in abstract.
Text p5,r143-147 (ref 13) better be moved from results to introduction, as background information.
Earlier study (ref 13), revealed heterozygous carriers of 3.58% and the present study (3 groups) revealed carriers for: 10000 controls 2.21%, PG 2.6% and CG 2.85%. Please give statistical methods used for group comparison as well as statistical results, even if no significance. The authors draw conclusion that the result of no difference between the groups is due to study limitation but an alternative conclusion is that there in fact was no difference between the groups in this population and that it was representative for a bigger population. The results in this study (but maybe other studies) does not support the statement on page 6, r211-3 of clinical significance of heterozygous carriage in cancer development. In future studies an improved male/female ratio and focus on gastrointestinal tumors with previously reported increased ratio in CF, should be aimed for.
I agree that more studies are warranted but have difficulties to see how the present study and the present knowledge should alter counseling for sporadic and hereditary cancers (as written in abstract). Limited word count in abstract and it is better to include the above mentioned part 3 in the study with 100000 healthy subjects.
It is difficult to compare the results of the present studies with discussed studies where OR for cancer groups/control groups was given but not percentage. The percentage carriers in the population is expected to be the same as percentage carries for cancer groups.
Ref 20 needs correction, it is from 2008, not 2018
Author Response
Thank you very much for your revision
I have with interest read this study. See below some suggestions for clarification.
Abstract: add that in addition to cancer and control group, WGS was performed in 10000 healthy individuals, add results in abstract.
Done.
Text p5,r143-147 (ref 13) better be moved from results to introduction, as background information.
Supplemented and transferred to the discussion.
Earlier study (ref 13), revealed heterozygous carriers of 3.58% and the present study (3 groups) revealed carriers for: 10000 controls 2.21%, PG 2.6% and CG 2.85%. Please give statistical methods used for group comparison as well as statistical results, even if no significance.
Done.
The authors draw conclusion that the result of no difference between the groups is due to study limitation but an alternative conclusion is that there in fact was no difference between the groups in this population and that it was representative for a bigger population. The results in this study (but maybe other studies) does not support the statement on page 6, r211-3 of clinical significance of heterozygous carriage in cancer development. In future studies an improved male/female ratio and focus on gastrointestinal tumors with previously reported increased ratio in CF, should be aimed for.
Corrected the paragraph
I agree that more studies are warranted but have difficulties to see how the present study and the present knowledge should alter counseling for sporadic and hereditary cancers (as written in abstract).
Deleted «This may lead to the emergence of a new therapeutic target, as well as to improved counseling for patients with sporadic and hereditary cancers.» and «Understanding the role of heterozygous genetic variants in CFTR gene may lead to the emergence of a new therapeutic target, as well as to improved counseling for patients with sporadic cancers and HCS.»
Limited word count in abstract and it is better to include the above mentioned part 3 in the study with 100000 healthy subjects.
Done.
It is difficult to compare the results of the present studies with discussed studies where OR for cancer groups/control groups was given but not percentage. The percentage carriers in the population is expected to be the same as percentage carries for cancer groups.
Added OR.
Ref 20 needs correction, it is from 2008, not 2018
Done.
Round 2
Reviewer 1 Report
Majority of concerns are addressed.